# Post-acute sequelae of SARS-CoV-2 with clinical condition definitions and comparison in a matched cohort

Michael A. Horberg [1,5] ✉, Eric Watson[1,5], Mamta Bhatia[1,5], Celeena Jefferson[1,5], Julia M. Certa[2,5], Seohyun Kim [1,5], Lily Fathi[1,5], Keri N. Althoff[3,5], Carolyn Williams[4,5] & Richard Moore[3,5]

Disease characterization of Post-Acute Sequelae of SARS-CoV-2 (PASC) does not account for pre-existing conditions and time course of incidence. We utilized longitudinal data and matching to a COVID PCR-negative population to discriminate PASC conditions over time within our patient population during 2020. Clinical Classification Software was used to identify PASC condition groupings. Conditions were specified acute and persistent (occurring 0-30 days post COVID PCR and persisted 30–120 days post-test) or late (occurring initially 30-120 days post-test). We matched 3:1 COVID PCR-negative COVIDPCR-positive by age, sex, testing month and service area, controlling for pre-existing conditions up to four years prior; 28,118 PCR-positive to 70,293 PCR-negative patients resulted. We estimated PASC risk from the matched cohort. Risk of any PASC condition was 12% greater for PCR-positive patients in the late period with a significantly higher risk of anosmia, cardiac dysrhythmia, diabetes, genitourinary disorders, malaise, and non-specific chest pain. Our findings contribute to a more refined PASC definition which can enhance clinical care.

SARS-CoV-2, the viral cause of COVID-19, has triggered a global pandemic infecting over 536 million people and over 6.3 million deaths worldwide as of June 2022[1]. The acute effects of COVID-19 are well documented, but longer-term effects are actively being investigated and defined. Post-acute sequelae of SARS-CoV-2 infection (PASC) has been described by the World Health Organization as the persistence of symptoms or new symptoms more than 30 days post-SARS-CoV-2 infection[2–4]. Lingering symptoms can persist months after acute infection, including recurring fatigue, muscle weakness, dyspnea, anxiety, and depression. The symptomatology of PASC and time course, however, have been based on self-selected cases and not drawn from a well-defined PCR-positive population, compared to an appropriate control group and followed longitudinally.

Disease characterization and definition have changed over time and identification via standard ICD-10 diagnosis codes were only enacted in later 2021[5]. Al-Aly and colleagues laid the groundwork by performing a comprehensive analysis of potential PASC symptomology within the Veterans Health Administration (VHA)[3]. Estiri et al. expanded PASC investigation to a non-hospitalized based cohort and identified 33 phenotypes in 3-6 month and 6-9-month periods post-COVID[6]. However, questions remain regarding the timing of conditions, and importantly what symptoms persisted from acute infection to late periods and which symptoms developed in the late period. Additionally, comparisons to matched populations with a negative SARS-CoV-2 test result (PCR-negative) have not been systematically conducted and are crucial to differentiating the impact of the

[1]Kaiser Permanente Mid-Atlantic Permanente Medical Group, Mid-Atlantic Permanente Research Institute, Rockville, MD 20852, USA. [2]United Health Group, Frederick, MD 21703, USA. [3]Johns Hopkins University, Baltimore, MD 21205, USA. [4]National Institute of Allergy and Infectious Diseases, National Institute of Health, Rockville, MD 20892, USA. [5]These authors contributed equally: Michael A. Horberg, Eric Watson, Mamta Bhatia, Celeena Jefferson, Julia M. Certa, Seohyun Kim, Lily Fathi, Keri N. Althoff, Carolyn Williams, Richard Moore. ✉e-mail: Michael.Horberg@kp.org

**Table 1 | Positive SARS-CoV-2 RT-PCR patient demographics**

| Category | Positive SARS-CoV-2 RT-PCR Patient Population |
|---|---|
| **Total, n (%)** | |
| Total | 31,390 (100.0%) |
| **Sex, n (%)** | |
| Female | 17,631 (56.2%) |
| Male | 13,759 (43.8%) |
| **Age (in years), n (%)** | |
| 18-29 | 6279 (20.0%) |
| 30-49 | 12,401 (39.5%) |
| 50-64 | 9014 (28.7%) |
| 65+ | 3696 (11.8%) |
| **Race/Ethnicity (self-reported), n (%)** | |
| Asian | 3191 (10.2%) |
| Black | 12,120 (38.6%) |
| Hispanic | 9044 (28.8%) |
| White | 5425 (17.3%) |
| **BMI (kg/m², n (%)** | |
| 25-29.9 (Overweight) | 7252 (23.1%) |
| 30-39.9 (Obesity) | 9042 (28.8%) |
| 40 + (Severe Obesity) | 2591 (8.3%) |
| **Comorbidities in pre-existing conditions period, n (%)** | |
| Chronic Kidney Disease | 921 (2.9%) |
| Chronic Obstructive Pulmonary Disease (COPD) | 282 (0.9%) |
| Diabetes | 5998 (19.1%) |
| HIV | 260 (0.8%) |
| Pregnancy | 569 (1.8%) |
| Malignancy | 790 (2.5%) |

**Table 2 | Clinical Classification Software (CCS) - PASC-related conditions deemed clinically significant by our infectious disease physicians among PCR-positive patients**

| CCS PASC Related Conditions | Time Period | | |
|---|---|---|---|
| | Late | Acute and persistent | Pre-existing conditions |
| Other lower respiratory disease | 2.73% | 4.50% | 10.40% |
| Diabetes | 2.25% | 1.32% | 9.85% |
| Gastrointestinal Disease | 4.19% | 1.59% | 2.89% |
| Conditions associated with dizziness or vertigo | 4.06% | 0.91% | 3.06% |
| Abdominal pain | 4.34% | 0.71% | 2.95% |
| Nonspecific chest pain | 4.17% | 1.42% | 1.61% |
| Mental health | 2.64% | 0.61% | 3.21% |
| Anxiety disorders | 2.74% | 0.86% | 2.27% |
| Genitourinary symptoms and ill-defined conditions | 3.45% | 0.46% | 1.78% |
| Malaise and fatigue | 3.42% | 1.02% | 0.84% |
| Cardiac dysrhythmias | 2.26% | 1.22% | 1.31% |
| Other nervous system disorders | 1.75% | 0.46% | 1.04% |
| Respiratory failure, insufficiency, arrest | 0.21% | 2.69% | 0.23% |
| Nausea and vomiting | 1.66% | 0.30% | 0.54% |
| Fluid and electrolyte disorders | 0.99% | 0.85% | 0.55% |
| Other nutritional, endocrine, and metabolic disorders | 1.16% | 0.17% | 0.31% |
| Anosmia | 0.73% | 0.05% | 0.01% |

Time periods were defined as follows: Late: 30-120 days post-PCR-positive test date; Acute and persistent 0-30 days post-PCR-positive test date and persisted 30-120 days; Pre-existing conditions: four years prior to PCR-positive test date

pandemic from the impact of viral infection. A case-control approach permits a better curated PASC definition and condition identification.

Our primary aim is to define a set of PASC conditions, and to describe the timing of the conditions, by applying to a diverse population and comparison group of similar PCR-negative individuals. We selected a longitudinal cohort of COVIDPCR-positive patients and matched them to COVIDPCR-negative patients within the Kaiser Permanente Mid-Atlantic States (KPMAS), identified the clinical conditions for which there is an increased risk for those PCR-positive (vs. PCR-negative) and estimated PASC incidence among those PCR-positive.

## Results

31,390 total PCR-positive patients were identified. The majority were female and over half were less than 50 years old (Table 1). Over half were minority populations with 39% Black and 29% Hispanic. Over half were overweight (BMI > 25 kg/m²). The most frequent pre-existing conditions co-morbidity was diabetes mellitus.

### PASC-related conditions among PCR-positive patients

We identified 17 PASC-related conditions (Table 2). The most common acute and persistent PASC-related conditions, that were either greater than the pre-existing conditions time interval or determined clinically significant during physician review, were other lower respiratory disease (4.5%) and respiratory failure (2.7%). Most common late PASC-related conditions (i.e., >1.5% among PCR-positive) were abdominal pain, gastrointestinal disorders, other nervous system disorders, nausea and vomiting, nonspecific chest pain, dizziness/vertigo, malaise and fatigue, anxiety disorders, mental health disorders, other lower

respiratory diseases, and cardiac dysrhythmias. Overall, 37.7% of PCR-positive patients had at least one condition (in the acute and persistent or late period) and 16.5% of PCR-positive patients had at least one PASC-related condition in either period. 20.4% of our PCR-positive patients had at least one condition and 4.1% had a PASC-related condition in the acute and persistent period. Late period results were 26.1% and 13.6%, respectively.

### Matching

The scaled matching algorithm resulted in a study population of 28,118 PCR-positive and 70,293 PCR-negative. 1:3 case to control matching represented 66.8% of the identified cohort, followed by 16.2% with 1:2 and 16.8% with 1:1 matching. Overall, both case and control groups had ~57% female patients, a higher distribution of Black (~40–43%) and Hispanic (~20–24%) compared with white (~18–22%) patients, ~87% distribution less than 65 years old, and 30%-33% distribution of patients with a BMI ≥ 30 kg/m² (Table 3). Although Chi-Squared statistics showed association between cohort and age, BMI, COPD, hospitalization in the 30 -120 daytime period post $T_0$, pregnancy, and race, all those associations were extremely weak (Highest Cramer's V = 0.060) and likely an effect of overall cohort size. Overall, 37.7% of PCR-positive patients had at least one condition (in the acute and persistent or late period) and 16.5% of PCR-positive patients had at least one PASC-related condition in either period. 20.4% of our PCR-positive patients had at least one condition and 4.1% had a PASC-related condition in the acute and persistent period. Late period results were 26.1% and 13.6%, respectively. Among PCR-negative, 2.5% had a PASC-related condition in the acute period and 22.1% had any condition in this period (more than the PCR-positive). Further, among

**Table 3 | Patient demographics and co-morbidities for the matched cohort**

| | Case (COVID PCR (+)) | Control (COVID PCR (-)) | Chi² P Value[a] | Cramer's V Value[b] |
|---|---|---|---|---|
| **Total, n (%)** | | | | |
| Total | 28,118 (100.0%) | 70,293 (100.0%) | | |
| **Sex, n (%)** | | | 0.091 | 0.005 |
| Female | 15,993 (56.9%) | 40,396 (57.5%) | | |
| Male | 12,125 (43.1%) | 29,897 (42.5%) | | |
| **Age in years, n (%)** | | | 0.0015 | 0.015 |
| 18–29 | 5791 (20.6%) | 14,805 (21.1%) | | |
| 30–39 | 5624 (20.0%) | 14,028 (20.0%) | | |
| 40–49 | 5352 (19.0%) | 12,828 (18.2%) | | |
| 50–64 | 7948 (28.3%) | 19,543 (27.8%) | | |
| 65–74 | 2449 (8.7%) | 6516 (9.3%) | | |
| 75–84 | 775 (2.8%) | 2109 (3.0%) | | |
| 85+ | 179 (0.6%) | 464 (0.7%) | | |
| **Race/Ethnicity (self-reported), n (%)** | | | <0.0001 | 0.051 |
| Asian | 3018 (10.7%) | 7772 (11.1%) | | |
| Black | 11,402 (40.6%) | 29,680 (42.2%) | | |
| Hispanic | 6877 (24.5%) | 14,057 (20.0%) | | |
| Unknown | 1496 (5.3%) | 3821 (5.4%) | | |
| White | 5325 (18.9%) | 14,963 (21.3%) | | |
| **Service area, n (%)** | | | 0.8019 | 0.003 |
| Baltimore | 5199 (18.5%) | 13,177 (18.7%) | | |
| DC and Southern Maryland | 13,337 (47.4%) | 33,247 (47.3%) | | |
| Northern Virginia | 9578 (34.1%) | 23,861 (33.9%) | | |
| Unknown | 4 (0.0%) | 8 (0.0%) | | |
| **BMI (kg/m²), n (%)** | | | <0.0001 | 0.060 |
| <18.5 (Underweight) | 146 (0.5%) | 597 (0.8%) | | |
| 18.5-24.9 (Healthy Weight) | 3750 (13.3%) | 12,179 (17.3%) | | |
| 25-29.9 (Overweight) | 6454 (23.0%) | 16,680 (23.7%) | | |
| 30-39.9 (Obesity) | 7956 (28.3%) | 17,916 (25.5%) | | |
| 40 + (Severe Obesity) | 2325 (8.3%) | 4781 (6.8%) | | |
| No Result | 7487 (26.6%) | 18,140 (25.8%) | | |
| **Co-morbidities, n (%)** | | | | |
| Chronic Kidney Disease | 849 (3.0%) | 2285 (3.3%) | 0.062 | 0.006 |
| COPD | 266 (0.9%) | 935 (1.3%) | <0.0001 | 0.016 |
| Diabetes Mellitus | 5271 (18.7%) | 11,498 (16.4%) | <0.0001 | 0.029 |
| Hepatitis B | 200 (0.7%) | 532 (0.8%) | 0.4525 | 0.002 |
| HIV | 239 (0.8%) | 567 (0.8%) | 0.4953 | 0.002 |
| Pregnancy | 489 (1.7%) | 1486 (2.1%) | 0.0002 | 0.012 |
| Malignancy | 729 (2.6%) | 2598 (3.7%) | <0.0001 | 0.028 |
| **Hospitalization[c], n (%)** | | | 0.1919 | 0.0042 |
| Hospitalization | 511 (1.8%) | 1366 (1.9%) | | |
| **Death Post Index[d], n (%)** | | | 0.0029 | 0.0095 |
| Death | 42 (0.1%) | 58 (0.1%) | | |

[a]p Values compare PCR-positive case patients to PCR-negative control patients for the specified demographic/co-morbidity. Analyses were performed using Pearson's chi-squared with α = 0.05. [b]Cramer's V results are based on absolute value. Values < 0.1 represents little to no association with the test groups. [c]Hospitalization represents if a patient was hospitalized in the 30–120 time period post index date. [d]Death represents if a patient died 0–120 days post index.

PCR-negative, 12.1% had a PASC-related condition in the late period, and 25.2% had any condition (fewer than the PCR-positive).

## Risk analysis – CCS PASC-related conditions

There was an increased risk of numerous conditions in those PCR-positive compared to PCR-negative in both acute and persistent and late time periods (Table 4; Fig. 1). The risk of having any conditions in the late period was 4% higher in PCR-positive versus PCR-negative (RR = 1.04; 95%CI: 1.01,1.07) and 8% lower in the acute and persistent period (RR = 0.92; 0.89,0.95). The risk of having at least one PASC-related condition, however, was increased by 12% in the late period (RR = 1.12; 1.08,1.16) and 60% in the acute and persistent period (RR = 1.60; 1.48,1.72). These PASC related conditions had significantly higher risk among PCR-positive versus PCR-negative in the late period: anosmia (RR = 3.88; 2.79,5.40); cardiac dysrhythmias (RR = 1.25; 1.08,1.45); diabetes (RR = 1.20; 1.03,1.38); genitourinary conditions (RR = 1.21; 1.07,1.36); malaise and fatigue (RR = 1.60; 1.41,1.81), non-specific chest pain (RR = 1.39; 1.24,1.55). For risk and cumulative incidence of all CCS categories considered, see Supplementary Table 1.

Some of these conditions also had increased risk among PCR-positive (vs. PCR-negative) in the acute and persistent period, including cardiac dysrhythmias (RR = 1.90; 95%CI: 1.45, 2.49), diabetes (RR = 1.96; 1.50, 2.55), malaise and fatigue (RR = 2.89; 2.10, 3.98), non-specific chest pain (RR = 2.39; 1.85, 3.10). Additional PASC related conditions that had increased risk in the acute and persistent period include other lower respiratory disease (RR = 2.51; 2.15, 2.92) and respiratory failure/insufficiency/arrest (RR = 22.95; 14.78, 35.64).

## Distribution analysis for PASC-related conditions

There was some variation in the demographic distributions for those experiencing at least one PASC-related condition in the acute and persistent and/or late periods (Table 5; Fig. 1). Most notably, those experiencing a PASC-related condition, versus the overall cohorts, were mostly female (~62% vs ~57%), slightly older (Age 65 + : ~15% vs 12–13%) and had higher hospitalization in the 30–120 days post lab test date (~5.2–6.8% vs 1.8–1.9%).

## Sensitivity analysis

The sensitivity analysis attenuated the increased risk ratios for PASC related conditions by allowing visible conditions to occur in multiple time periods; thus, removing the requirement of mutual exclusivity between time periods. None of the significant increased risk ratios changed statistical significance (Table 6). Abdominal pain (RR = 0.73; 95%CI: 0.63,0.85) and nausea and vomiting (RR = 0.61; 0.45,0.82) had protective associations in the acute and persistent period that strengthened and were significant in the sensitivity analysis; both conditions were less burdensome among PCR-positive and PCR-negative groups.

The case/control diabetes and corticosteroid sensitivity analysis showed no significant association between cases and controls and corticosteroid use during the late time period (P = 0.89) had a significant, but weak, association (P = 0.01; Cramer's V = 0.1169), between cases and controls and corticosteroid use during the acute and persistent time period. Only 32% of diabetic patients (cases and controls) were on corticosteroids during the late period and only 18% of diabetic patients (cases and controls) were on corticosteroids during the acute and persistent time period. The risk ratio for diabetes in the late time period was 1.20 (CI: 1.03, 1.38) and 1.96 (CI: 1.50, 2.55) in the acute and persistent time period; therefore, we can say that corticosteroid use and/or abuse likely has little to no association to the increased diabetes risk.

## Discussion

Our study introduces a list of clinical conditions associated with PASC and an overall incidence of PASC within our population that can be

## Table 4 | Risk and cumulative incidence of CCS categories in the case vs. controls

| CCS Condition with Time Period | Case Cumulative Incidence | Control Cumulative Incidence | Risk Ratio [95% CI] |
|---|---|---|---|
| **Total - Any condition** | | | |
| Late | 26.1% | 25.2% | 1.04 [1.01,1.07] * |
| Acute and persistent | 20.4% | 22.1% | 0.92 [0.89,0.95] * |
| Pre-existing conditions | 46.1% | 44.2% | 1.04 [1.02,1.06] * |
| **Total - PASC related conditions** | | | |
| Late | 13.6% | 12.1% | 1.12 [1.08,1.16] * |
| Acute and persistent | 4.1% | 2.5% | 1.60 [1.48,1.72] * |
| Pre-existing conditions | 30.7% | 29.0% | 1.06 [1.03,1.08] * |
| **Abdominal pain** | | | |
| Late | 1.8% | 1.7% | 1.05 [0.95,1.17] |
| Acute and persistent | 0.2% | 0.2% | 0.87 [0.63,1.20] |
| Pre-existing conditions | 3.4% | 3.4% | 1.00 [0.93,1.07] |
| **Anosmia** | | | |
| Late | 0.3% | 0.1% | 3.88 [2.79,5.40] * |
| Acute and persistent | 0.0% | 0.0% | 0.50 [0.11,2.28] |
| Pre-existing conditions | 0.0% | 0.0% | 1.25 [0.31,5.00] |
| **Anxiety disorders** | | | |
| Late | 1.1% | 1.1% | 1.01 [0.89,1.15] |
| Acute and persistent | 0.2% | 0.3% | 0.83 [0.63,1.10] |
| Pre-existing conditions | 2.8% | 3.6% | 0.78 [0.72,0.84] * |
| **Cardiac dysrhythmias** | | | |
| Late | 0.9% | 0.7% | 1.25 [1.08,1.45] * |
| Acute and persistent | 0.3% | 0.2% | 1.90 [1.45,2.49] * |
| Pre-existing conditions | 1.6% | 1.5% | 1.02 [0.91,1.14] |
| **Conditions associated with dizziness or vertigo** | | | |
| Late | 1.7% | 1.6% | 1.05 [0.94,1.16] |
| Acute and persistent | 0.2% | 0.2% | 1.08 [0.82,1.44] |
| Pre-existing conditions | 3.6% | 3.4% | 1.04 [0.96,1.12] |
| **Diabetes** | | | |
| Late | 0.9% | 0.8% | 1.20 [1.03,1.38] * |
| Acute and persistent | 0.3% | 0.2% | 1.96 [1.50,2.55] * |
| Pre-existing conditions | 11.5% | 9.3% | 1.23 [1.18,1.29] * |
| **Fluid and electrolyte disorders** | | | |
| Late | 0.4% | 0.6% | 0.73 [0.59,0.90] * |
| Acute and persistent | 0.2% | 0.1% | 1.96 [1.41,2.74] * |
| Pre-existing conditions | 0.7% | 0.8% | 0.82 [0.69,0.97] * |
| **Genitourinary symptoms and ill-defined conditions** | | | |
| Late | 1.5% | 1.2% | 1.21 [1.07,1.36] * |
| Acute and persistent | 0.1% | 0.1% | 1.14 [0.75,1.74] |
| Pre-existing conditions | 2.1% | 2.0% | 1.04 [0.94,1.14] |
| **Gastrointestinal Disease** | | | |
| Late | 1.7% | 1.7% | 1.00 [0.90,1.12] |

## Table 4 (continued) | Risk and cumulative incidence of CCS categories in the case vs. controls

| CCS Condition with Time Period | Case Cumulative Incidence | Control Cumulative Incidence | Risk Ratio [95% CI] |
|---|---|---|---|
| Acute and persistent | 0.4% | 0.4% | 0.98 [0.79,1.21] |
| Pre-existing conditions | 3.4% | 4.0% | 0.84 [0.78,0.90] * |
| **Malaise and fatigue** | | | |
| Late | 1.4% | 0.9% | 1.60 [1.41,1.81] * |
| Acute and persistent | 0.3% | 0.1% | 2.89 [2.10,3.98] * |
| Pre-existing conditions | 1.0% | 0.9% | 1.15 [1.00,1.33] * |
| **Mental health** | | | |
| Late | 1.1% | 1.2% | 0.95 [0.83,1.08] |
| Acute and persistent | 0.2% | 0.3% | 0.62 [0.45,0.86] * |
| Pre-existing conditions | 4.0% | 5.6% | 0.71 [0.67,0.76] * |
| **Nausea and vomiting** | | | |
| Late | 0.7% | 0.7% | 0.95 [0.80,1.12] |
| Acute and persistent | 0.1% | 0.1% | 0.81 [0.49,1.32] |
| Pre-existing conditions | 0.6% | 0.7% | 0.84 [0.71,1.00] |
| **Nonspecific chest pain** | | | |
| Late | 1.7% | 1.2% | 1.39 [1.24,1.55] * |
| Acute and persistent | 0.4% | 0.2% | 2.39 [1.85,3.10] * |
| Pre-existing conditions | 1.8% | 1.5% | 1.25 [1.12,1.39] * |
| **Other lower respiratory disease** | | | |
| Late | 1.1% | 1.2% | 0.92 [0.80,1.04] |
| Acute and persistent | 1.2% | 0.5% | 2.51 [2.15,2.92] * |
| Pre-existing conditions | 12.2% | 10.0% | 1.21 [1.16,1.26] * |
| **Other nervous system disorders** | | | |
| Late | 0.7% | 0.7% | 1.04 [0.89,1.22] |
| Acute and persistent | 0.1% | 0.1% | 0.97 [0.64,1.49] |
| Pre-existing conditions | 1.3% | 1.5% | 0.87 [0.77,0.98] * |
| **Other nutritional; endocrine; and metabolic disorders** | | | |
| Late | 0.5% | 0.4% | 1.14 [0.93,1.40] |
| Acute and persistent | 0.0% | 0.1% | 0.54 [0.30,0.99] * |
| Pre-existing conditions | 0.4% | 0.4% | 0.82 [0.66,1.03] |
| **Respiratory failure; insufficiency; arrest (adult)** | | | |
| Late | 0.1% | 0.1% | 1.14 [0.73,1.80] |
| Acute and persistent | 0.7% | 0.0% | 22.95 [14.78,35.64] * |
| Pre-existing conditions | 0.3% | 0.2% | 1.72 [1.29,2.29] * |

* Represents risk ratios with $p < 0.05$: Time periods were defined as follows: Late: 30–120 days post COVID test date; Acute and persistent 0-30 days post COVID test date and persisted 30-120 days; Pre-existing conditions: four years prior to COVID test date. All risk ratios presented are unadjusted.

used to diagnose long-term effects of COVID-19. Additionally, our results aid in characterizing an operational PASC definition and provide a time frame for identifying conditions with significantly higher incidence post-COVID-19 infection, including those described by others[7]. Existing literature hasn't fully explored PASC and long-term

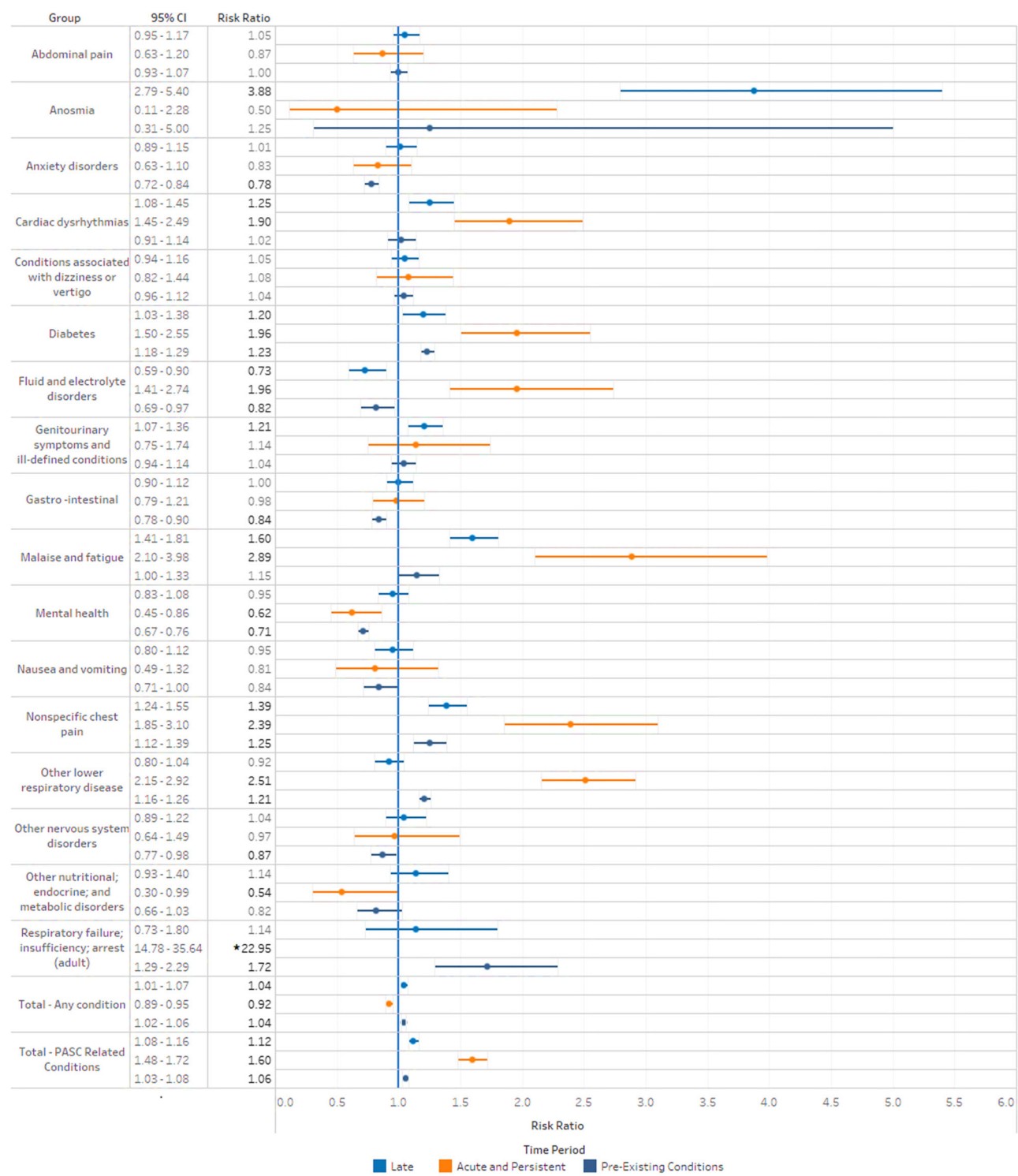

**Fig. 1 | Unadjusted risk ratios (and 95% confidence intervals) of PASC-related conditions comparing PCR-positive (vs. PCR-negative), in three-time periods anchored on the date of SARS-CoV-2 PCR test result.** A CCS condition risk ratio comparison with a 95% CI plot for PCR-positive population vs PCR-negative population within our study time periods. Risk ratio is the measure of interest comprised of the number of CCS conditions incident in the PCR-positive cohort ($n = 28,118$) versus CCS conditions incident in the PCR-negative cohort ($n = 70,293$), with 95% confidence intervals represented by the respective bands. Utilizing 1.0 as the baseline, significant risk ratios ($p < 0.05$) for the PASC-related conditions can be identified in bold and compared in scale to the other conditions. (*) Asterisk designates that the metric was too large to fit within the scale of the graphic.

effects of COVID-19 infection[8]. We expand upon previous studies by comparing PCR-positive patients to a matched cohort of PCR-negative patients within a closed integrated health system and utilized a time-based approach to provide supporting evidence for the resulting common clinical conditions of PASC.

Unlike other studies, we separated conditions by time of presentation and accounted for previous pre-existing conditions in order to delineate conditions of PASC. While many pre-existing conditions may have been exacerbated by COVID, operationally, they should not be considered a late PASC condition. Future research should aim to

**Table 5 | Patient demographics for the matched cohort vs patients experiencing PASC related conditions**

| | Case (COVID PCR (+) with PASC[a] | Total case (COVID PCR (+)) | Control (COVID PCR (-)) with PASC[b] | Total control (COVID PCR (-)) |
|---|---|---|---|---|
| **Total, n (%)** | | | | |
| Total | 4625 (100.0%) | 28,118 (100.0%) | 9745 (100.0%) | 70,293 (100.0%) |
| **Sex, n (%)** | | | | |
| Female | 2866 (62.0%) | 15,993 (56.9%) | 6092 (62.5%) | 40,396 (57.5%) |
| Male | 1759 (38.0%) | 12,125 (43.1%) | 3653 (37.5%) | 29,897 (42.5%) |
| **Age in years, n (%)** | | | | |
| 18–29 | 866 (18.7%) | 5791 (20.6%) | 1985 (20.4%) | 14,805 (21.1%) |
| 30–39 | 899 (19.4%) | 5624 (20.0%) | 1898 (19.5%) | 14,028 (20.0%) |
| 40–49 | 810 (17.5%) | 5352 (19.0%) | 1695 (17.4%) | 12,828 (18.2%) |
| 50–64 | 1355 (29.3%) | 7948 (28.3%) | 2693 (27.6%) | 19,543 (27.8%) |
| 65–74 | 480 (10.4%) | 2449 (8.7%) | 1018 (10.4%) | 6516 (9.3%) |
| 75–84 | 178 (3.8%) | 775 (2.8%) | 363 (3.7%) | 2109 (3.0%) |
| 85+ | 37 (0.8%) | 179 (0.6%) | 93 (1.0%) | 464 (0.7%) |
| **Race/Ethnicity (self-reported), n (%)** | | | | |
| Asian | 465 (10.1%) | 3018 (10.7%) | 977 (10.0%) | 7772 (11.1%) |
| Black | 1903 (41.1%) | 11,402 (40.6%) | 4266 (43.8%) | 29,680 (42.2%) |
| Hispanic | 1189 (25.7%) | 6877 (24.5%) | 2012 (20.6%) | 14,057 (20.0%) |
| White | 877 (19.0%) | 5325 (18.9%) | 2041 (20.9%) | 14,963 (21.3%) |
| Unknown | 191 (4.1%) | 1496 (5.3%) | 449 (4.6%) | 3821 (5.4%) |
| **Service area, n (%)** | | | | |
| Baltimore | 907 (19.6%) | 5199 (18.5%) | 1987 (20.4%) | 13,177 (18.7%) |
| DC and Southern Maryland | 2127 (46.0%) | 13,337 (47.4%) | 4536 (46.5%) | 33,247 (47.3%) |
| Northern Virginia | 1590 (34.4%) | 9578 (34.1%) | 3222 (33.1%) | 23,861 (33.9%) |
| Unknown | 1 (0.0%) | 4 (0.0%) | 0 (0.0%) | 8 (0.0%) |
| **BMI, n (%)** | | | | |
| <18.5 (Underweight) | 22 (0.5%) | 146 (0.5%) | 85 (0.9%) | 597 (0.8%) |
| 18.5–24.9 (Healthy Weight) | 601 (13.0%) | 3750 (13.3%) | 1639 (16.8%) | 12,179 (17.3%) |
| 25–29.9 (Overweight) | 1043 (22.6%) | 6454 (23.0%) | 2331 (23.9%) | 16,680 (23.7%) |
| 30–39.9 (Obesity) | 1418 (30.7%) | 7956 (28.3%) | 2624 (26.9%) | 17,916 (25.5%) |
| 40 + (Severe Obesity) | 468 (10.1%) | 2325 (8.3%) | 802 (8.2%) | 4781 (6.8%) |
| No Result | 1073 (23.2%) | 7487 (26.6%) | 2264 (23.2%) | 18,140 (25.8%) |
| **Co-morbidities, n (%)** | | | | |
| Chronic Kidney Disease | 179 (3.9%) | 849 (3.0%) | 402 (4.1%) | 2285 (3.3%) |
| COPD | 45 (1.0%) | 266 (0.9%) | 182 (1.9%) | 935 (1.3%) |
| Diabetes Mellitus | 981 (21.2%) | 5271 (18.7%) | 1756 (18.0%) | 11,498 (16.4%) |
| Hepatitis B | 31 (0.7%) | 200 (0.7%) | 75 (0.8%) | 532 (0.8%) |
| HIV | 50 (1.1%) | 239 (0.8%) | 92 (0.9%) | 567 (0.8%) |
| Pregnancy | 126 (2.7%) | 489 (1.7%) | 310 (3.2%) | 1486 (2.1%) |
| Malignancy | 140 (3.0%) | 729 (2.6%) | 485 (5.0%) | 2598 (3.7%) |
| **Hospitalization[c], n (%)** | | | | |
| Hospitalization | 239 (5.2%) | 511 (1.8%) | 665 (6.8%) | 1366 (1.9%) |
| **Death Post Index[d], n (%)** | | | | |
| Death | 8 (0.2%) | 42 (0.1%) | 30 (0.3%) | 58 (0.1%) |

[a]COVID PCR( + ) patients that experienced at least one PASC related condition in the acute and persistent and/or late periods. [b]COVID PCR(-) patients that experienced at least one PASC related condition in the acute and persistent and/or late periods. [c]Hospitalization represents if a patient was hospitalized in the 30–120-time period post index date. [d]Death represents if a patient died in the 0-120 days post index.

understand the severity and persistence of these PASC-related conditions. Additional time periods post-COVID would be important in analyzing additional conditions or symptoms that develop well beyond expected time intervals, including PASC development with later surges/waves of COVID-19 and the impact of vaccination on PASC.

We limited our cohort to this initial period of SARS-CoV-2 infection (i.e., 2020) to avoid the influence of later variants and vaccinations, and to only those with a PCR test result[9].

Our study also reveals a presence of a disease burden among PCR-negative persons. While any conditions and our PASC-related conditions are distinctly different, the comparison of any condition provides additional evidence that the resulting PASC-related conditions are not only higher risk in the PCR-positive group, but higher risk compared to patients experiencing other symptoms/conditions. This comparison reenforces that the resulting PASC-related conditions are truly representative of patients experiencing PASC. Also, note that among patients without COVID, many experienced these same symptoms, providing further context for our results.

In contrast, the absolute differences in PASC-related conditions, or any conditions, are not large when comparing PCR-positive to PCR-negative groups. In fact, any condition (having any condition that was considered for PASC in our final analysis) was more common among PCR-negative than PCR-positive in the acute period. This has further implications for an operational PASC definition, in that while many conditions have been cited as potentially part of PASC, they are occurring with similar frequencies among PCR-negative patients. It is important, thus, to recognize that these symptoms, while elevated in patients with SARS-CoV-2 infection, are not unusual either in PCR-negative persons. Further, it is important to acknowledge the toll the pandemic has taken on all patients and while many of these PASC-defining conditions do not have large incidence rates, these conditions are still very impactful to the patients that experience them and require attention from their medical providers.

Our acute and persistent and late period PASC-related conditions are not surprising, as most have been described in case reports in the literature to date or are commonly seen in sub-acute viral illnesses[10,11]. It also should be noted that we did not compare our results or rates to a separate viral condition, such as HIV, as COVID and PASC were unique to this time and the focus of our study. While these PASC condition categories are multifaceted, such as GU disorders, all have been described by others[12,13]. Respiratory symptoms were more prominent in the acute and persistent period, which is consistent with COVID-19 symptomatology[14]. However, most of these did not occur in the late period, and many patients had pre-existing pulmonary conditions and respiratory-related diagnoses. As described in the literature, anosmia was seen at a higher incidence for PCR-positive patients[7].

An acknowledged limitation is that not all conditions described in the literature or popular press can be coded consistently during our study period in the EHR, most notably brain fog; however, malaise and mental health were prominent PASC-related conditions and likely associated with brain fog symptomatology[15]. Consistent with the findings from Al-Aly[3], diabetes mellitus had increased incidence during the post-COVID period. One possibility is that diabetic patients were simply undiagnosed until they sought care for their COVID-19 infection and were laterally diagnosed[16,17]. Another possibility is that COVID affects blood sugar and pancreatic endocrine function[17]. As noted in our sensitivity analysis, corticosteroid use did not greatly impact diabetes incidence. However, diabetes is relatively common in the KPMAS population, and an even further increased risk of the disease is of considerable concern for patient health. Future research is needed to understand the relationship between diabetes and COVID-19.

Al-Aly and colleagues[3] provided an encompassing view of PASC conditions within the VHA population. Our study provides supporting evidence around specific conditions identified and negates their limitations around having a primarily older (average 60 years), white and male population[3]. We also utilized a time interval analysis which provides context and support for our most profound PASC-related conditions. Our results differ from the VHA study as they have an overall higher level of risk for most of their identified conditions. One potential reason for this difference is that our control group required

**Table 6 | Sensitivity analysis - risk and cumulative incidence of symptom-based CCS categories**

| CCS Category and Time Period | Mutually Exclusive[a] | Case Cumulative Incidence | Control Cumulative Incidence | Risk Ratio [95% CI] |
|---|---|---|---|---|
| **Abdominal pain** | | | | |
| Late | Yes | 1.8% | 1.7% | 1.05 [0.95,1.17] |
| Late | No | 5.4% | 5.3% | 1.01 [0.95,1.07] |
| Pre-existing conditions | Yes | 0.2% | 0.2% | 0.87 [0.63,1.20] |
| Pre-existing conditions | No | 0.8% | 1.1% | 0.73 [0.63,0.85] * |
| **Anosmia** | | | | |
| Late | Yes | 0.3% | 0.1% | 3.88 [2.79,5.40] * |
| Late | No | 0.3% | 0.1% | 3.21 [2.37,4.35] * |
| Pre-existing conditions | Yes | 0.0% | 0.0% | 0.50 [0.11,2.28] |
| Pre-existing conditions | No | 0.0% | 0.0% | 0.33 [0.08,1.46] |
| **Conditions associated with dizziness or vertigo** | | | | |
| Late | Yes | 1.7% | 1.6% | 1.05 [0.94,1.16] |
| Late | No | 5.5% | 5.3% | 1.04 [0.98,1.11] |
| Pre-existing conditions | Yes | 0.2% | 0.2% | 1.08 [0.82,1.44] |
| Pre-existing conditions | No | 1.1% | 1.1% | 1.02 [0.89,1.16] |
| **Malaise and fatigue** | | | | |
| Late | Yes | 1.4% | 0.9% | 1.60 [1.41,1.81] * |
| Late | No | 2.7% | 1.9% | 1.46 [1.33,1.60] * |
| Pre-existing conditions | Yes | 0.3% | 0.1% | 2.89 [2.10,3.98] * |
| Pre-existing conditions | No | 0.6% | 0.3% | 2.07 [1.69,2.54] * |
| **Nausea and vomiting** | | | | |
| Late | Yes | 0.7% | 0.7% | 0.95 [0.80,1.12] |
| Late | No | 1.4% | 1.5% | 0.90 [0.80,1.01] |
| Pre-existing conditions | Yes | 0.1% | 0.1% | 0.81 [0.49,1.32] |
| Pre-existing conditions | No | 0.2% | 0.3% | 0.61 [0.45,0.82] * |
| **Nonspecific chest pain** | | | | |
| Late | Yes | 1.7% | 1.2% | 1.39 [1.24,1.55] * |
| Late | No | 3.9% | 2.8% | 1.38 [1.28,1.49] * |
| Pre-existing conditions | Yes | 0.4% | 0.2% | 2.39 [1.85,3.10] * |
| Pre-existing conditions | No | 0.9% | 0.5% | 1.84 [1.56,2.16] * |

[a]Mutually exclusive Yes represents our original analysis where pre-existing conditions were removed from calculating acute and persistent and late counts. Mutually exclusive No represents the sensitivity analysis where we allowed pre-existing conditions to be counted in the acute and persistent and late time periods. * Represents risk ratios with $p < 0.05$

testing negative for COVID-19, while their control group includes those who had no evidence of testing. Demographic differences may have also contributed to the result divergence. Our study also improves upon Estiri and colleagues' PCR-negative comparison group by utilizing data from a closed integrated healthcare system with accurate membership accounting, applying a matching algorithm to better control for confounding variables, and most importantly, providing additional comparison periods to provide supporting evidence for the late conditions[6].

Other limitations are relevant to our study. It is possible that pre-existing conditions could be found more than four years prior to the PCR test date. The intent of going back four years was to ensure that we capture conditions that may have been missed on more recent encounters prior to the COVID test. Individual risk estimates are also heavily dependent on time period length, by which larger time periods, such as the 4-year pre-existing condition period, likely have a high probability of diagnosis capture compared to the 30-day acute and persistent period. This time period length discrepancy has been attenuated in our analysis as our overall comparisons between the PCR-negative vs PCR-positive cohorts are compared within each equitable time interval. Additionally, our sensitivity analysis found no effect on the significance of our results when removing the mutual exclusivity requirement for time periods and allowing symptom-based conditions to count in the acute and persistent period as well as the late period, regardless of if a condition was pre-existing.

Additionally, KPMAS healthcare utilization patterns were also found to be significantly altered by the pandemic[18]. While we capture all telehealth visits, there is the potential for missing PASC diagnoses and encounters as some patients may not have sought medical care for their symptoms. We tested this limitation and found >76% of our PCR-positive and PCR-negative cohorts had at least one encounter, virtual or in-person, in the late period. Lastly, we acknowledge that our current study period does not include the outside influence of other variants, vaccination, and widely distributed home testing which may impact future definitions and symptomatology of PASC.

Our study population consists of insured patients only which includes Medicare and Medicaid, and charity care, so a wide demographic is included. While geographically limited, our population represents the general population well in the DC/VA/MD area[4,5]. We also cannot rule out the potential for missing data around PCR testing, especially in early 2020. Diagnoses, lab results and death, which was not compared with the National Death Index (NDI) for the cause of death, also have the potential for missingness; however, we believe our care model and connection to external data sources significantly reduces much of this limitation. Lastly, we cannot rule out the possibility of false negative COVID tests but given the high community prevalence of SARS-CoV-2 during the study period, and the KPMAS protocol for testing 5-14 days post COVID-19 exposure, the likelihood of missed COVID-19 diagnosis is low. Our strengths include examining

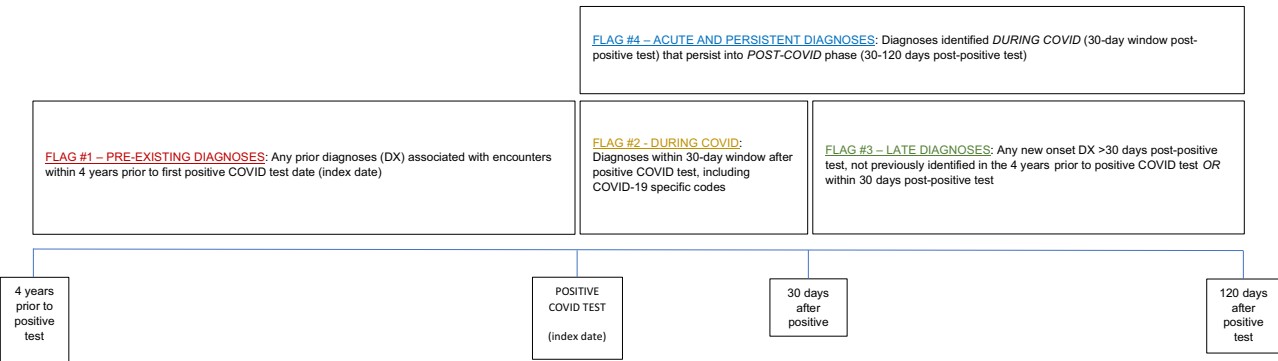

**Fig. 2 | Diagnosis Observation Periods.** Diagnostic observation timeline for CCS conditions in relation to the COVID testing date as the index date. The time periods used in this study were defined as follows: Late: 30–120 days post COVID test date; Acute and persistent 0–30 days post COVID test date and persisted 30–120 days; Pre-existing conditions: four years prior to COVID test date.

data within an integrated and closed health system and drawing from a well-defined patient population of over 800,000 members. Further, our system has comprehensive capture of both PCR-positive and PCR-negative test results from both internal and external sources, as well as a comprehensive capture of PASC recorded symptoms and conditions. Additionally, our ability to create a matched PCR-negative population, with majority of PCR-positive cases being matched to three COVID controls, and our analysis of distinct time periods associated with condition manifestation are key distinctions of this study compared to others in the literature[16].

Our study demonstrates a clearly defined set of conditions for PASC definition and delineation within an integrated care system. This delineation compared the acute and persistent time period conditions with conditions identified in the late time period. These conditions are at a significantly higher risk when compared with a PCR-negative population matched on similar demographics. However, PASC-related conditions do occur among PCR-negative populations and should not be neglected among these patients. Additionally, our study found that the overall cumulative incidence of PASC, as defined by COVID positive patients with a PASC-related diagnosis in the acute and persistent or late periods, is 16.5%. Importantly, the low-risk levels, defined by the cumulative incidence of each individual condition, provide context to the overall low burden of disease for PASC-related conditions in the KPMAS population. These findings contribute to the overall evaluation of PASC and can be employed by clinicians in their care of patients who are diagnosed with COVID-19. Our research provides supporting evidence for an accepted operational definition for PASC; however, understanding of the severity and duration of these conditions will be crucial.

## Methods
### Setting
Kaiser Permanente (KP) is an integrated health system in the United States, with over 800,000 members in the Mid-Atlantic region, representing Maryland, the District of Columbia, and Northern Virginia. KPMAS members are a diverse population and their demographics represent their respective jurisdictions[19]. They are provided comprehensive integrated health care, including, but not limited to, primary and specialty care, ambulatory and inpatient care (with integration among partner hospitals in the Mid-Atlantic region). Their healthcare is coordinated through an integrated electronic health record (EHR) system which includes clinical data, financial information (claims data) on services received external to KPMAS, and data from the Geographically Enriched Member Sociodemographic (GEMS) database[20]. KPMAS is a closed healthcare system with high ascertainment of COVID-19 in the population, as well as potential PASC conditions and symptoms.

Our study was approved by the KPMAS Institutional Review Board on an expedited basis.

### Study population and COVID-19 classification
SARS-CoV-2 RT-PCR (PCR) testing, the most widely available test during our study period, has been regarded as the gold standard for COVID patient identification[21]. Given the magnitude of testing performed within our system and external testing linkages, we classified COVID positive patients as those with a confirmed PCR result and refer to those as PCR-positive. We refer to those with only COVID PCR-negative results as PCR-negative.

Utilizing KPMAS EHR, including internal and external records incorporated into the EHR (Epic® Care Everywhere and Maryland/Washington DC health information exchange called CRISP)[9], we identified adult patients (≥18 years) who had a PCR result between January 1, 2020, through December 31, 2020. We limited our cohort to this period to avoid the influence of later variants and vaccinations, and only to those with a PCR test result[9]. Of note, the KPMAS protocol did not test patients prior to five days post-exposure or greater than 14 days post-exposure. We prioritized PCR-positive results for each patient, then selected the first positive date as our index date. Patients were classified into cases when they received a PCR-positive COVID result or controls if they only tested negative. We excluded patients not enrolled in KPMAS 120 days post-PCR test date.

For each PCR-positive patient (case), we matched up to three PCR-negative patients, without replacement, by PCR testing month and year (using their first negative test date), age group at the time of PCR test (18–29, 30–39, 40–49, 50–64, 65–74, 75–84, ≥85 years), race/ethnicity (Black/African American, White, Hispanic, Asian, Other/Unknown), sex (female or male), and service area (to account for any physician differences in diagnostic practices). When 1:3 matching was not possible, cases were matched to controls 1:2 or 1:1.

Demographic covariates of interest abstracted from the EHR included: race/ethnicity age, comorbidities (chronic kidney disease, chronic obstructive pulmonary disease, diabetes mellitus type 1 or 2, Hepatitis B, HIV, cancer), Body Mass Index (BMI; kg/m²), insurance type (commercial, Medicare, Medicaid, Charity, Affordable Care Act), pregnancy status, service area, and hospitalizations (30-120 days post-test) and known deaths post-index date.

### Timing and classification of symptoms and conditions
The timing and definition of the conditions identified were critical in distinguishing sequelae of significance. Our index date ($T_0$) was the PCR test date. Three, mutually exclusive, diagnostic time intervals were identified and anchored on $T_0$: (1) pre-existing conditions time interval - diagnoses up to four years prior to $T_0$, (2) acute and persistent time interval – diagnoses occurring 0–30 days post-$T_0$ and

**Table 7 | CCS categories merged in analysis**

| ORIGINAL CCS CATEGORY | ORIGINAL CCS CATEGORY DESCRIPTION | MERGED CCS CATEGORY DESCRIPTION |
|---|---|---|
| 124 | Acute and chronic tonsillitis | Infectious disease |
| 157 | Acute and unspecified renal failure | Renal |
| 125 | Acute bronchitis | Other lower respiratory disease |
| 109 | Acute cerebrovascular disease | Vascular Disease & CVD |
| 100 | Acute myocardial infarction | Vascular Disease & CVD |
| 660 | Alcohol-related disorders | Mental health |
| 147 | Anal and rectal conditions | Hemorrhoids |
| 116 | Aortic and peripheral arterial embolism or thrombosis | Vascular Disease & CVD |
| 115 | Aortic; peripheral; and visceral artery aneurysms | Vascular Disease & CVD |
| 142 | Appendicitis and other appendiceal conditions | GI |
| 128 | Asthma | Other lower respiratory disease |
| 652 | Attention-deficit conduct and disruptive behavior disorders | Mental health |
| 3 | Bacterial infection; unspecified site | Infectious disease |
| 149 | Biliary tract disease | Other liver disease |
| 21 | Cancer of bone and connective tissue | Cancer |
| 35 | Cancer of brain and nervous system | Cancer |
| 24 | Cancer of breast | Cancer |
| 19 | Cancer of bronchus; lung | Cancer |
| 26 | Cancer of cervix | Cancer |
| 14 | Cancer of colon | Cancer |
| 33 | Cancer of kidney and renal pelvis | Cancer |
| 16 | Cancer of liver and intrahepatic bile duct | Cancer |
| 18 | Cancer of other GI organs; peritoneum | Cancer |
| 27 | Cancer of ovary | Cancer |
| 29 | Cancer of prostate | Cancer |
| 15 | Cancer of rectum and anus | Cancer |
| 13 | Cancer of stomach | Cancer |
| 41 | Cancer; other and unspecified primary | Cancer |
| 158 | Chronic kidney disease | Renal |
| 127 | Chronic obstructive pulmonary disease and bronchiectasis | Other lower respiratory disease |
| 108 | Congestive heart failure; non-hypertensive | Vascular Disease & CVD |
| 101 | Coronary atherosclerosis and other heart disease | Vascular Disease & CVD |
| 653 | Delirium dementia and amnestic and other cognitive disorders | Mental health |
| 50 | Diabetes mellitus with complications | Diabetes |
| 49 | Diabetes mellitus without complication | Diabetes |
| 137 | Diseases of mouth; excluding dental | Infectious disease |
| 98 | Essential hypertension | Hypertension |
| 246 | Fever of unknown origin | Infectious disease |
| 248 | Gangrene | Infectious disease |
| 140 | Gastritis and duodenitis | GI |
| 139 | Gastroduodenal ulcer (except hemorrhage) | GI |
| 153 | Gastrointestinal hemorrhage | GI |
| 88 | Glaucoma | Eye |
| 54 | Gout and other crystal arthropathies | Joint disease |
| 84 | Headache; including migraine | Conditions associated with dizziness or vertigo |
| 96 | Heart valve disorders | Vascular Disease & CVD |
| 6 | Hepatitis | Other liver diseases |
| 5 | HIV infection | Infectious disease |
| 99 | Hypertension with complications and secondary hypertension | Hypertension |
| 656 | Impulse control disorders NEC | Mental health |
| 201 | Infective arthritis and osteomyelitis (except that caused by tuberculosis or sexually transmitted disease) | Infectious disease |
| 123 | Influenza | Infectious disease |
| 135 | Intestinal infection | Infectious disease |

**Table 7 (continued) | CCS categories merged in analysis**

| ORIGINAL CCS CATEGORY | ORIGINAL CCS CATEGORY DESCRIPTION | MERGED CCS CATEGORY DESCRIPTION |
|---|---|---|
| 43 | Malignant neoplasm without specification of site | Cancer |
| 670 | Miscellaneous mental health disorders | Mental health |
| 657 | Mood disorders | Mental health |
| 40 | Multiple myeloma | Cancer |
| 156 | Nephritis; nephrosis; renal sclerosis | Renal |
| 38 | Non-Hodgkin's lymphoma | Cancer |
| 154 | Noninfectious gastroenteritis | GI |
| 110 | Occlusion or stenosis of precerebral arteries | Vascular Disease & CVD |
| 203 | Osteoarthritis | Joint disease |
| 111 | Other and ill-defined cerebrovascular disease | Vascular Disease & CVD |
| 104 | Other and ill-defined heart disease | Vascular Disease & CVD |
| 212 | Other bone disease and musculoskeletal deformities | Joint disease |
| 78 | Other CNS infection and poliomyelitis | Infectious disease |
| 162 | Other diseases of bladder and urethra | Renal |
| 161 | Other diseases of kidney and ureters | Renal |
| 121 | Other diseases of veins and lymphatics | Vascular Disease & CVD |
| 141 | Other disorders of stomach and duodenum | GI |
| 155 | Other gastrointestinal disorders | GI |
| 198 | Other inflammatory condition of skin | Other skin disorders |
| 23 | Other nonepithelial cancer of skin | Cancer |
| 134 | Other upper respiratory disease | Other lower respiratory disease |
| 126 | Other upper respiratory infections | Other lower respiratory disease |
| 152 | Pancreatic disorders (not diabetes) | GI |
| 97 | Peri-; endo-; and myocarditis; cardiomyopathy (except that caused by tuberculosis or sexually transmitted disease) | Vascular Disease & CVD |
| 118 | Phlebitis; thrombophlebitis and thromboembolism | Vascular Disease & CVD |
| 130 | Pleurisy; pneumothorax; pulmonary collapse | Other lower respiratory disease |
| 122 | Pneumonia (except that caused by tuberculosis or sexually transmitted disease) | Other lower respiratory disease |
| 103 | Pulmonary heart disease | Other lower respiratory disease |
| 87 | Retinal detachments; defects; vascular occlusion; and retinopathy | Eye |
| 659 | Schizophrenia and other psychotic disorders | Mental health |
| 2 | Septicemia (except in labor) | Infectious disease |
| 197 | Skin and subcutaneous tissue infections | Other skin disorders |
| 661 | Substance-related disorders | Mental health |
| 662 | Suicide and intentional self-inflicted injury | Mental health |
| 245 | Syncope | Conditions associated with dizziness or vertigo |
| 112 | Transient cerebral ischemia | Vascular Disease & CVD |
| 1 | Tuberculosis | Infectious disease |
| 119 | Varicose veins of lower extremity | Vascular Disease & CVD |
| 7 | Viral infection | Infectious disease |

CCS Categories merged based on clinical determination that it was appropriate to merge to other CCS categories or a separate CCS Category needed to be created

persisted into the 30–120 days period, but not previously identified in the pre-existing conditions time interval, and (3) late time interval - new disease diagnoses 30–120 days post-$T_0$, but not previously identified in the pre-existing conditions or acute and persistent time intervals (Fig. 2).

Diagnostic grouping of ICD codes was performed with standard Healthcare Cost and Utilization Project (HCUP) Clinical Classifications Software (CCS)[22]. From this software, the CCS Category Level was chosen as the anchor group as it allowed for general diagnostic rollup but maintained enough specificity to identify distinct conditions for PASC. Further modification of the CCS condition mapping was performed after review by two KPMAS infectious disease physicians. It was determined that some ICD code mappings, for example, anosmia, did not meet expectation and were either excluded from the CCS

mapping, remapped to another CCS Category, or placed under a CCS Category that we created (Tables 7, 8; excluded diagnoses/CCS categories Tables Supplementary Table 2-Supplementary Table 3). This method of manual modifications to CCS Categories has been performed in previous studies[23]. We abstracted all diagnoses from our EHR and claims systems that occurred within our observation periods and enforced mutual exclusivity time requirements at the CCS Category Level. CCS conditions were only counted once per patient and classified based on when the condition was first recorded in the EHR.

**Distribution Analysis and PASC-related conditions**
To determine which CCS conditions provided the signal for PASC diagnosis, distributions were calculated for PCR-positive patients by taking the distinct number of patients with a particular CCS condition

**Table 8 | Specific ICD diagnoses merged in analysis**

| ORIGINAL CCS CATEGORY | ORIGINAL CCS CATEGORY DESCRIPTION | ICD 10 CM CODE | ICD 10 CM CODE DESCRIPTION | MERGED CCS CATEGORY DESCRIPTION |
|---|---|---|---|---|
| 95 | Other nervous system disorders | R200 | Anesthesia of skin | Skin Sensitivity |
| 95 | Other nervous system disorders | R430 | Anosmia | Anosmia |
| 95 | Other nervous system disorders | G5603 | Carpal tunnel syndrome, bilateral upper limbs | Neuropathy |
| 95 | Other nervous system disorders | G5602 | Carpal tunnel syndrome, left upper limb | Neuropathy |
| 95 | Other nervous system disorders | G5601 | Carpal tunnel syndrome, right upper limb | Neuropathy |
| 95 | Other nervous system disorders | G5600 | Carpal tunnel syndrome, unspecified upper limb | Neuropathy |
| 95 | Other nervous system disorders | R203 | Hyperesthesia | Skin Sensitivity |
| 95 | Other nervous system disorders | R201 | Hypoesthesia of skin | Skin Sensitivity |
| 95 | Other nervous system disorders | G5632 | Lesion of radial nerve, left upper limb | Neuropathy |
| 95 | Other nervous system disorders | G5631 | Lesion of radial nerve, right upper limb | Neuropathy |
| 95 | Other nervous system disorders | G5623 | Lesion of ulnar nerve, bilateral upper limbs | Neuropathy |
| 95 | Other nervous system disorders | G5622 | Lesion of ulnar nerve, left upper limb | Neuropathy |
| 95 | Other nervous system disorders | G5621 | Lesion of ulnar nerve, right upper limb | Neuropathy |
| 95 | Other nervous system disorders | R208 | Other disturbances of skin sensation | Skin Sensitivity |
| 95 | Other nervous system disorders | R438 | Other disturbances of smell and taste | Anosmia |
| 95 | Other nervous system disorders | G5612 | Other lesions of median nerve, left upper limb | Neuropathy |
| 95 | Other nervous system disorders | R432 | Parageusia | Anosmia |
| 95 | Other nervous system disorders | R202 | Paresthesia of skin | Skin Sensitivity |
| 95 | Other nervous system disorders | R431 | Parosmia | Anosmia |
| 95 | Other nervous system disorders | R209 | Unspecified disturbances of skin sensation | Skin Sensitivity |
| 95 | Other nervous system disorders | R439 | Unspecified disturbances of smell and taste | Anosmia |
| 95 | Other nervous system disorders | G5693 | Unspecified mononeuropathy of bilateral upper limbs | Neuropathy |
| 95 | Other nervous system disorders | G5691 | Unspecified mononeuropathy of right upper limb | Neuropathy |
| 58 | Other nutritional; endocrine; and metabolic disorders | E807 | Disorder of bilirubin metabolism, unspecified | GI |
| 58 | Other nutritional; endocrine; and metabolic disorders | E839 | Disorder of mineral metabolism, unspecified | Diabetes |
| 58 | Other nutritional; endocrine; and metabolic disorders | E801 | Porphyria cutanea tarda | Other hematologic conditions |
| 259 | Residual codes; unclassified | R69 | Illness, unspecified | General Symptoms and Illness |
| 259 | Residual codes; unclassified | R6889 | Other general symptoms and signs | General Symptoms and Illness |
| 259 | Residual codes; unclassified | Z7289 | Other problems related to lifestyle | Mental health |

ICD diagnoses merged based on clinical determination that it was appropriate to merge to other CCS categories or a separate CCS Category needed to be created

over the total number of distinct patient-CCS condition combinations, within each respective time interval. We used an aggregated total distribution percentage, summed between all time periods, of 0.04% as a cutoff for CCS Conditions that merited clinical review. The .04% cutoff was determined by review of distribution counts for the CCS conditions, whereby the .04% cutoff merited an appropriate number of conditions for risk analysis. From the remaining symptoms and diagnoses, higher frequency conditions were reviewed by two KPMAS infectious disease physicians. Conditions flagged by the infection disease physicians, based on biologic plausibility and review of the medical literature to date of initial analysis (April 2021), were then further refined and grouped on clinical similarities and/or a more defined condition classification. For example, Genitourinary symptoms were grouped together while CCS conditions related to trauma were deleted from the analysis. These condition groupings were again presented to the KPMAS infectious disease physicians with a final determination made to which conditions had a plausible biologic association to PASC

(Supplementary Table 1). CCS conditions that met our criteria of higher acute and persistent or late distributions, and deemed clinically significant by the infectious disease physicians, were considered PASC-related conditions (PASC-related conditions; Table 2).

**Statistical analysis**
Demographic characteristics were compared by COVID status. Slight variations in demographic characteristics used for matching (caused by the scaled matching technique) were tested via Cramer's V to investigate distribution equality in cases and controls. Further distribution analyses were performed on those experiencing at least one PASC-related condition in the acute and persistent and/or late periods. Overall counts, cumulative incidence, and unadjusted risk ratios with 95% confidence intervals (using the Wald Test method) were calculated for each CCS condition category within each time interval. Cumulative incidence was defined as the total number of distinct patients with a particular CCS condition, within each respective time

period, over the total number of patients in the observed cohort. Additionally, totals for a patient having at least one CCS condition or at least one PASC-related condition were estimated and stratified by time interval. Data collection and analyses were performed using SAS software (version 9.4; Cary, North Carolina), SQL developer (version 17.3.1) and Tableau (version 2019.1). A p-value <0.05 guided statistical interpretation.

## Sensitivity analysis

To mitigate a potential limiting effect of our CCS selection criteria, we performed a sensitivity analysis on the PASC-related conditions deemed symptoms. CCS condition counts for abdominal pain, anosmia, conditions associated with dizziness/vertigo, malaise and fatigue, nausea and vomiting, nonspecific chest pain were recalculated by removing the pre-existing conditions diagnosis exclusion requirement for identification of acute and persistent and late diagnoses, thus allowing the presence of these symptoms to be counted irrespectively. Statistical analysis was performed in the same manner as our primary analysis.

In addition, to account for the potential effects of the use and/or abuse of corticosteroids on glucose levels, we performed a sensitivity analysis on patients identified as having diabetes mellitus in the acute and persistent or late periods. Dispensed corticosteroid medications during the time periods in question were identified for those patients. A chi-squared test was used to determine if an association was present between case and control patients with a diabetic diagnosis and corticosteroid use.

## Reporting summary

Further information on research design is available in the Nature Research Reporting Summary linked to this article.

## Data availability

For privacy and legal requirements, data is only available upon request which must comply with the Mid-Atlantic Permanente Research Institute (MAPRI) security, privacy, and intellectual property standards. All data requests will be reviewed by the MAPRI compliance officer and principal investigator within 30 business days upon receipt of request. All requesters will be required to provide a statement of need and comply with MAPRI policy and requirements. Failure to produce sufficient information and/or a request that doesn't meet MAPRI policy, may be denied. All data were collected from internal Kaiser Permanente Mid-Atlantic States databases that are utilized for clinical care and claims. Information on the structure and name of the tables used in these databases are proprietary and cannot be shared publicly.

## Code availability

For privacy and legal requirements, project related code is only available upon request which must comply with the Mid-Atlantic Permanente Research Institute (MAPRI) security, privacy, and intellectual property standards. All code requests will be reviewed by the MAPRI compliance officer and principal investigator within 30 business days upon receipt of request. All requesters will be required to provide a statement of need and comply with MAPRI policy and requirements. Failure to produce sufficient information and/or a request that doesn't meet MAPRI policy, may be denied. All data were collected from internal Kaiser Permanente Mid-Atlantic States databases that are utilized for clinical care and claims. Information on the structure and name of the tables used in these databases are proprietary and cannot be shared publicly.

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

## Acknowledgements
The content is solely the responsibility of the authors and does not necessarily represent the official views of the National Institutes of Health. This work was supported by National Institutes of Health grants U01AI069918, F31AI124794, F31DA037788, G12MD007583, K01AI093197, K01AI131895, K23EY013707, K24AI065298, K24AI118591, K24DA000432, KL2TR000421, N01CP01004, N02CP055504, N02CP91027, P30AI027757, P30AI027763, P30AI027767, P30AI036219, P30AI050409, P30AI050410, P30AI094189, P30AI110527, P30MH62246, R01AA016893, R01DA011602, R01DA012568, R01 AG053100, R24AI067039, U01AA013566, U01AA020790, U01AI038855, U01AI038858, U01AI068634, U01AI068636, U01AI069432, U01AI069434, U01DA03629, U01DA036935, U10EY008057, U10EY008052, U10EY008067, U01HL146192, U01HL146193, U01HL146194, U01HL146201, U01HL146202, U01HL146203, U01HL146204, U01HL146205, U01HL146208, U01HL146240, U01HL146241, U01HL146242, U01HL146245, U01HL146333, U24AA020794,U54MD007587, UL1RR024131, UL1TR000004, UL1TR000083, Z01CP010214 and Z01CP010176; contracts CDC-200-2006-18797 and CDC-200-2015-63931 from the Centers for Disease Control and Prevention, USA; contract 90047713 from the Agency for Healthcare Research and Quality, USA; contract 90051652 from the Health Resources and Services Administration, USA; grants CBR-86906, CBR-94036, HCP-97105 and TGF-96118 from the Canadian Institutes of Health Research, Canada; Ontario Ministry of Health and Long Term Care; and the Government of Alberta, Canada. Additional support was provided by the National Institute Of Allergy And Infectious Diseases (NIAID), National Cancer Institute (NCI), National Heart, Lung, and Blood Institute (NHLBI), Eunice Kennedy Shriver National Institute Of Child Health & Human Development (NICHD), National Human Genome Research Institute (NHGRI), National Institute for Mental Health (NIMH) and National Institute on Drug Abuse (NIDA), National Institute On Aging (NIA), National Institute Of Dental & Craniofacial Research (NIDCR), National Institute Of Neurological Disorders And Stroke (NINDS), National Institute Of Nursing Research (NINR), National Institute on Alcohol Abuse and Alcoholism (NIAAA), National Institute on Deafness and Other Communication Disorders (NIDCD), and National Institute of Diabetes and Digestive and Kidney Diseases (NIDDK). Authors M.H., E.W., M.B., C.J., J.C., S.K., L.F., K.A., and R.M. were awarded funding as part of the North American AIDS Cohort Collaboration on Research and Design (NA-ACCORD) supplemental grant.

## Author contributions
M.H., E.W., C.J., J.C., K.A., C.W., and R.M. designed the research. M.H., E.W., C.J., S.K., and R.M. conducted data collections and analyses. M.H., E.W., M.B., C.J., J.C., S.K., L.F., K.A., C.W., and R.M. participated in manuscript writing, editing and submission.

## Competing interests
The authors declare no competing interests.
