## [Peer Review File · Nature Communications]

Post-acute sequelae of SARS-CoV-2 with clinical condition definitions and comparison in a matched cohortEditorial Note: This manuscript has been previously reviewed at another journal that is not operating a transparent peer review scheme. This document only contains reviewer comments and rebuttal letters for versions considered at *Nature Communications*.

Reviewer #4 (Remarks to the Author):

This is an interesting paper. However, it has a few shortcomings.

First, it is unhelpful to give an overall estimate of PASC and 'Any conditions' as they are so diverse conditions. Please remove these analyses.

Second, the data for this study is from the early period of the pandemic prior to the vaccines and based on the original variant. (Wuhan variant). However, the symptoms have changed since. For example anosmia was one of the key symptoms for the original variant but this is no longer the case for individuals infected with omicron.

Third, I am concerned about your findings related to diabetes. As you suggest in the discussion it is likely that diabetic patients were simply undiagnosed until they sought care for their COVID-19 infection and were laterally diagnosed. In other words it is likely that those who tested positive were likely to undergo further investigations compared to those who tested negative. The same may apply to cardiac dysrhythmias and genitourinary symptoms.

Overall, I would suggest the paper is rewritten and focus on the fact that there are very few conditions that are persistent after COVID infections.

Reviewer #5 (Remarks to the Author):

Thank you for the opportunity to review this well-written study describing relative risk of PASC conditions in the post-COVID period. I do think that several other recent papers describe using large EHR databases to determine risk of various conditions post-COVID--but, I do think that the case-control match is novel, and the use of a closed health care system (KP) adds a new dimension as well. My comments follow:

A figure would be immensely helpful to illustrate the time periods (acute, late, etc. on a visual timeline). This would orient the reader for the Results section.

Line 187--typo, the period after race should be a comma.

Though it may have been true when this study was completed, I don't think line 237 ("Existing literature is limited regarding PASC and long-term effects of COVID-19 infection") is true any longer.

Line 243: While I do think it is useful to separate out exacerbated pre-existing conditions and new conditions for this analysis, I don't think one can unequivocally say that exacerbated pre-existing conditions "should not be considered a late PASC condition." For patients experiencing significant worsening of a prior condition specifically due to PASC, having that recognized is essential. Additionally, since you looked back 4 years for pre-existing conditions, it's highly possible that a patient might have had a condition (say, dyspnea) 4 years ago, which resolved, and then returned much later as PASC-related dyspnea. This strikes me as quite different than a chronic condition that has no resolution, which may or may not exacerbate due to PASC.

Paragraph starting on line 250: It strikes me as difficult to make any firm conclusions about the PCR-negative group, as the only factor bringing them together is the absence of a particular disease (COVId). While it is certainly fair to say that the increase in absolute risk of PASC conditions was not large when compared with controls, stating that there is a "high burden of disease" among the PCR negative population may be unintentionally misleading.

Line 329: Would suggest removing the word "only" from "only 16.5%"--the significance of that number is very great on a personal level if one is part of that 16.5%. This is particularly important considering that this study does not account for the severity of the reported conditions, as the

authors state on line 334.

Assuming this is as true within KP as other US health care systems, I would add a limitation that it was immensely difficult to get a PCR test for the first part of 2020, meaning that case ascertainment is going to be spotty during that period, even at KP.

REVIEWER COMMENTS

Reviewer #4 (Remarks to the Author):

This is an interesting paper. However, it has a few shortcomings.

We thank you for providing your insights and opinions, and appreciate your praise and review of our study and manuscript. We hope that we have successfully addressed your concerns.

First, it is unhelpful to give an overall estimate of PASC and 'Any conditions' as they are so diverse conditions. Please remove these analyses.

Thank you for this comment and although we agree “Any Conditions” and our “PASC-related conditions” are distinctly different, we wanted to provide additional evidence (and comparison) that the resulting PASC-related conditions were not only higher risk in the COVID positive group, but higher risk compared to patients experiencing other symptoms/conditions. We believe this comparison substantiates our claim that the resulting PASC related conditions are truly representative of patients experiencing PASC. Also note that among patients without COVID, many experienced these same symptoms, putting our results in even further context. Please see additional lines 250-256.

Second, the data for this study is from the early period of the pandemic prior to the vaccines and based on the original variant. (Wuhan variant). However, the symptoms have changed since. For example, anosmia was one of the key symptoms for the original variant but this is no longer the case for individuals infected with omicron.

Thank you for this comment, and we agree that it is a limitation that later variants and other factors have not been analyzed in our study. We have added this limitation (line 313-315) to the manuscript. However, we do believe our time period analyzed is also a strength as vaccinations and home testing were not yet widely distributed (line 247-248) and this time period would have the most accurate data collected with less influence from these outside factors.

Third, I am concerned about your findings related to diabetes. As you suggest in the discussion it is likely that diabetic patients were simply undiagnosed until they sought care for their COVID-19 infection and were laterally diagnosed. In other words it is likely that those who tested positive were likely to undergo further investigations compared to those who tested negative. The same may apply to cardiac dysrhythmias and genitourinary symptoms.

We appreciate your concern and are also interested in the diabetes finding (corroborated by other studies). As an observational cohort analysis, we do not want to presume any explanation in advance, and discard potential associations of statistical, and potentially clinical, significance. As noted (line 132-134), KPMAS infectious disease physicians made the final determination to which resulting PASC related conditions had a plausible biologic association, of which diabetes was included. Clearly, more research is needed in this area and we added this statement to the manuscript (line 285). To note, we are planning to investigate this further in a future PASC study.

Overall, I would suggest the paper is rewritten and focus on the fact that there are very few conditions that are persistent after COVID infections.

Thank you for this comment. We agree this is a major finding of our paper, as we note in the methods

(accounting for this distinct period “acute and persistent” from “later and incident”), in our results section, and in the discussion section. Although we do agree that there is great significance in finding an overall low burden of disease for PASC related conditions (line 250-266), our key findings and analysis pertain to demonstrating a clearly defined set of conditions for PASC definition and delineation. We believe our study contributes significant findings in this area, for a unique and diverse population, and it will provide a foundation for future evaluations of PASC.

Reviewer #5 (Remarks to the Author):

Thank you for the opportunity to review this well-written study describing relative risk of PASC conditions in the post-COVID period. I do think that several other recent papers describe using large EHR databases to determine risk of various conditions post-COVID--but, I do think that the case-control match is novel, and the use of a closed health care system (KP) adds a new dimension as well. My comments follow:

A figure would be immensely helpful to illustrate the time periods (acute, late, etc. on a visual timeline). This would orient the reader for the Results section.

Thank you for your kind review and comment. We agree and have Figure 1 which indicates the observation periods. We emphasize the importance of our figure in a few places within the manuscript.

Line 187--typo, the period after race should be a comma.

Thank you, the period was replaced with comma.

Though it may have been true when this study was completed, I don't think line 237 ("Existing literature is limited regarding PASC and long-term effects of COVID-19 infection") is true any longer.

Thank you, we changed the wording to “existing literature hasn't fully explored PASC and long-term effects.”

Line 243: While I do think it is useful to separate out exacerbated pre-existing conditions and new conditions for this analysis, I don't think one can unequivocally say that exacerbated pre-existing conditions "should not be considered a late PASC condition." For patients experiencing significant worsening of a prior condition specifically due to PASC, having that recognized is essential. Additionally,

since you looked back 4 years for pre-existing conditions, it's highly possible that a patient might have had a condition (say, dyspnea) 4 years ago, which resolved, and then returned much later as PASC-related dyspnea. This strikes me as quite different than a chronic condition that has no resolution, which may or may not exacerbate due to PASC.

Thank you for your comment and although we agree that COVID and/or PASC can exacerbate pre-existing conditions, our primary objective was to delineate conditions of PASC to form a PASC definition. Therefore, we felt that the removal of pre-existing conditions was necessary in meeting our objective. Additionally, we performed a sensitivity analysis which removed the pre-existing exclusion and found no significant changes to our results (line 150 -156)

Paragraph starting on line 250: It strikes me as difficult to make any firm conclusions about the PCR-negative group, as the only factor bringing them together is the absence of a particular disease (COVID). While it is certainly fair to say that the increase in absolute risk of PASC conditions was not large when compared with controls, stating that there is a "high burden of disease" among the PCR negative population may be unintentionally misleading.

Thank you, we replaced "Our study also reveals a high burden of disease in PCR-negative persons" with "Our study also reveals a presence of a disease burden among PCR-negative persons." Please note that with other edits, this line is now numbered 250.

Line 329: Would suggest removing the word "only" from "only 16.5%"--the significance of that number is very great on a personal level if one is part of that 16.5%. This is particularly important considering that this study does not account for the severity of the reported conditions, as the authors state on line 334.

Thank you for pointing this out, and we removed the word "only" from line 339.

Assuming this is as true within KP as other US health care systems, I would add a limitation that it was immensely difficult to get a PCR test for the first part of 2020, meaning that case ascertainment is going to be spotty during that period, even at KP.

Thank you, we added to our limitation about missing data (line 309): "We also cannot rule out the potential for missing data around PCR testing, especially in early 2020. Diagnoses, lab results and death, which was not compared with the National Death Index (NDI) for cause of death, also have the potential for missingness; however, we believe our care model and connection to external data sources significantly reduces much of this limitation."